# Use of *Caenorhabditis elegans* to Unravel the Tripartite Interaction of Kynurenine Pathway, UPR^mt^ and Microbiome in Parkinson’s Disease

**DOI:** 10.3390/biom14111370

**Published:** 2024-10-28

**Authors:** Charles Viau, Alyssa Nouar, Jianguo Xia

**Affiliations:** 1Institute of Parasitology, McGill University, Sainte-Anne-de-Bellevue, QC H9X 3V9, Canada; charles.viau@mail.mcgill.ca (C.V.); alyssa.nouar@mail.mcgill.ca (A.N.); 2Department of Microbiology and Immunology, McGill University, Montreal, QC H3A 2B4, Canada

**Keywords:** *C. elegans*, microbiome, UPR^mt^, kynurenine, Parkinson’s disease

## Abstract

The model organism *Caenorhabditis elegans* and its relationship with the gut microbiome are gaining traction, especially for the study of neurodegenerative diseases such as Parkinson’s Disease (PD). Gut microbes are known to be able to alter kynurenine metabolites in the host, directly influencing innate immunity in *C. elegans*. While the mitochondrial unfolded protein response (UPR^mt^) was first characterized in *C. elegans* in 2007, its relevance in host–microbiome interactions has only become apparent in recent years. In this review, we provide novel insights into the current understanding of the microbiome–gut–brain axis with a focus on tripartite interactions between the UPR^mt^, kynurenine pathway, and microbiome in *C. elegans*, and explore their relationships for PD remediations.

## 1. Introduction

Parkinson’s disease (PD) is a neurodegenerative disease that is characterized by the death of dopamine-producing neurons in the substantia nigra region of the brain [1]. PD patients develop deficits such as tremors, rigidity, bradykinesia, and cognitive impairment [1]. Although the exact cause of PD remains unknown, researchers agree that manifestation of the disease is correlated to both genetic and environmental factors. Mutations in the *PRKN*, *SNCA*, *PINK1*, and *LRRK2* genes have been identified as potential risk factors, while environmental factors such as diet and gut microbiome are also found to be significantly associated with PD in human populations [2,3,4,5,6]. Indeed, some of the symptoms that precede and are present during disease manifestation include gastrointestinal abnormalities, and that the vagus nerve connecting between the enteric nervous system and the central nervous system may play a crucial role in PD pathogenesis [6,7]. Establishing a suitable animal model will be fundamental to elucidate the underlying molecular mechanisms for therapeutics development.

Since the characterization of its native gut microbiome in 2016, *Caenorhabditis elegans* has gained attention in recent years for being a useful model for the study of host–microbiome interactions [8,9,10]. The phylum-level composition in wild isolates of *C. elegans* were similar to the normal microbiome phyla of healthy humans, suggesting a conservation of microbiome functions between hosts, but in a simplified manner [9,10]. *C. elegans* do not have an adaptive immune system, although they have an innate immune system which shares similarity to those of humans. The worms have several signaling pathways that are conserved across species such as the p38 MAPK pathway and the DAF-2/DAF-16 insulin-like signaling pathway [11]. The nematodes also possess one Toll-like receptor (TLR) named TOL-1, and like its human counterpart, it also detects pathogen-associated molecular patterns (PAMPs) [12]. Moreover, *C. elegans* have an inducible antimicrobial response against certain pathogenic infection which is in part controlled by a TGF-ß pathway, again showing some conservation with human signaling pathways [13]. Hence, *C. elegans* serves as a model for studying gut microbiome interactions with the immune system in a simplified context, offering insights applicable to humans. *C. elegans* is an ideal model for revealing specific bacterial species, genes, and metabolites of the gut microbiome on worm physiology and fitness. *C. elegans* is a bacterivore, and some of the bacterial diets colonize the worm gut, as opposed to being digested and releasing small bioactive molecules. Hence, we can easily study the effect of the bacterial diet from either single (monoxenic) bacterial species, a mixture of species (cocktails), or the whole communities [8,9,10,14]. This technique allows for the illustration of complex interactions between worm host and microbe with one or multiple bacterial species at a time. For instance, using this method, it was shown that *Escherichia coli* OP50, the worm’s standard laboratory diet, altered the therapeutic effect of metformin (a type 2 diabetes medication) by modulating the worm’s folate/S-adenosylmethionine one carbon cycle [15]. More recently, various bacterial species have been studied with regard to their supplementation of vitamins and essential nutrients to the worm. For example, *Comamonas aquatica* DA1877 was found to accelerate development through increased vitamin B12 bioavailability [16,17]. Moreover, a combination of native microbiome members, *Comamonas* sp. MYb 131 and *Chryseobacterium* sp. CHNTR MYb 120, was found to increase vitamin B6 supply to the worm, compared to the standard laboratory diet of *E. coli* OP50, positively influencing worm fitness and extending their lifespan [14,18]. Overall, these reports have demonstrated that microbiome members can positively affect worm physiology and fitness. 

*C. elegans* is also well suited used for the study of neurodegenerative diseases such as PD [19,20]. Unlike the mammalian nervous system which consists of billions of neurons with incompletely understood synaptic connections, the nervous system of an adult *C. elegans* hermaphrodite consists of only 302 neurons that are completely mapped. The molecular and cellular functions of neurons are highly conserved between *C. elegans* and mammals, such as the presence of dopaminergic neurons [21]. In fact, α-synuclein has been successfully expressed in the body wall muscle cells and neurons of *C. elegans*. With this model, α-synuclein gradually forms aggregates in the cells where it is expressed, resulting in decreased worm motility over time, ultimately leading to death. Many reports have demonstrated the alleviation of the paralysis caused by this molecule through genetic means or by exposing the worms to therapeutic molecules [20,22]. However, some limitations of using *C. elegans* as a PD model is the lack of an intrinsic α-synuclein ortholog in this organism, as well as differential neuronal connectivity [23]. In humans, the etiology of PD results from the accumulation of α-synuclein protein aggregates in the substantia nigra compacta region of the brain, which causes the generation of Lewy bodies in dopaminergic neurons, ultimately leading to their destruction [24]. This aggregation leads to the induction of certain homeostatic events in the neuron, such as the activation of the mitochondrial unfolded protein response (UPR^mt^). Although α-synuclein was discovered over 25 years ago, targeting this protein and its aggregation in PD therapy remains challenging due to its multiple conformations and intrinsic role in neurotransmission [25,26]. Hence, it may be more effective to target the underlying causes of α-synuclein aggregation rather than the aggregates themselves.

The link between the gut microbiome and PD remains to be elucidated. Many reports have suggested that the two are intertwined, including chronic inflammation of the colon in PD patients [27]. For example, levels of short-chain fatty acids (SCFAs) (acetate, butyrate, and propionate) in the serum and feces of PD patients were shown to be lower than in healthy controls [28]. This observation was tied to the relative lower abundance of SCFA-generating anti-inflammatory bacteria in the feces of these patients, determined by 16S rRNA gene sequencing, such as *Blautia* and *Faecalibacterium*, whereas there was an increase in pro-inflammatory bacteria, such as *Bacteroides* and *Akkermansia*, compared to healthy controls [28]. The lower abundance of SCFAs in the intestine results in increased gut permeability, leading to the translocation of pro-inflammatory molecules such as lipopolysaccharide (LPS), leading to gut dysbiosis [29,30]. LPS has been shown to promote the aggregation of α-synuclein and to cause neuroinflammation via transport by the vagus nerve [31]. For example, the incidence of small intestine bacterial overgrowth (SIBO), a condition caused by gut dysbiosis, is positively associated with motor dysfunction, specifically in PD patients [31,32].

Here, we review the etiology of neurodegenerative diseases and more specifically, the influence of the microbiome on disease remediation using *C. elegans* as a model. Microbes, either colonizing or digested by the worm, have been shown to trigger UPR^mt^ and activate the kynurenine pathway, which induce worm innate immunity and potentially resolve PD etiology. Focusing on *C. elegans*, we discuss microbe activation of immune receptors, such as TOL-1, by bacterial pathogens or microbiome members, as well as UPR^mt^ activation and subversion by various bacterial species. We then explore how these interactions impact receptor-dependent innate immunity, host–microbiome dynamics, and the role of the kynurenine pathway in these processes. Finally, we discuss the potential of metabolomics in elucidating the influence of host–microbiome interactions with translational applications to human studies.

## 2. *C. elegans* as a Model for Host–Microbiome Interactions

### 2.1. Pathogen Recognition Receptors

*C. elegans* interacts with microbes through receptors. Pathogen recognition receptors (PRRs) such as mammalian Toll-like receptors (TLRs) play an integral role in this interaction as each subtype interacts with specific microbial ligands. TLRs can sense microbial-associated molecular patterns (MAMPs) or pathogen-associated molecular patterns (PAMPs) from the gut microbiome and ensure homeostasis by responding accordingly to these signals. In *C. elegans*, however, only one receptor of this class exists: TOL-1 [33]. Evidence that this receptor plays a role in pathogen and potentially microbiome interactions, has been documented. The full-length lipopolysaccharide (LPS) from *Salmonella enterica* serovar Typhimurium (*S*. Typhimurium) was able to activate TOL-1, compared to insignificant activation from the rough LPS *E. coli* OP50 strain (lacking the O-antigen) [33]. However, TOL-1 is only found in the pharynx and neurons, and not in the worm gut [12]. There is also evidence that the sole Toll-like receptor plays a role in pathogen avoidance [12]. Nonetheless, successful infection by *S*. Typhimurium involves proliferation in the *C. elegans* gut and eventual worm death, which is a premature phenotype observed in the *tol-1 C. elegans* knockdown or mutant [12,33]. Unlike TLR-4 in mammals, which only responds to LPS from Gram-negative bacteria, TOL-1 in *C. elegans* responds to both Gram-negative and Gram-positive bacteria, suggesting that there are additional ligands that this receptor senses [12]. For instance, it was found that small peptidoglycan muropeptides from *Enterococcus faecalis’* SagA enzyme activity promoted tolerance in *C. elegans* to *S*. Typhimurium in a TOL-1-dependent manner [34]. In brief, small muropeptides generated from *E. faecalis* peptidoglycan by this enzyme, through signalling by TOL-1, allow for *S*. Typhimurium persistence in the gut, which does not kill the worm prematurely. Although the identification of PRRs in *C. elegans* remains elusive, the downstream immune pathways resemble those of higher organisms, and it would be interesting to consider measuring TOL-1 activation with other bacterial species comprising the microbiome of the worm.

Upon infection by *S*. Typhimurium, TOL-1 is activated, leading to the downstream activation of innate immunity signaling through the adapter TRF-1 [12]. Unlike canonical mammalian TLRs, TOL-1 lacks a C-terminus extension after its cytoplasmic Toll-interleukin receptor (TIR) domain, suggesting that different innate immunity signaling pathways are regulated by this receptor [35]. It was shown that the TIR domain interacts with the p38 MAP kinase (PMK-1) [36], Figure 1. Induced factors, in response to *S*. Typhimurium infection, include the production of antimicrobial peptides—such as ABF-2, a defensin-like antimicrobial peptide—leading to the control of proliferation of the invading pathogen [12]. As mentioned previously, activation of TOL-1 also leads to pathogen avoidance behavior, given its localization in sensory neurons [12]. 

### 2.2. Additional Immune Receptors

There are other immune receptors in *C. elegans* induced by infection, such as C-type lectin-like domain receptors and G-protein coupled receptors. *C. elegans* encodes 283 C-type lectin-like domain receptors that are mostly expressed in the worm intestine, suggesting these proteins respond to a plethora of microbial ligands [37]. The downstream effectors of C-type lectin-like domain receptors also include PMK-1 [38]. Some C-type lectin-like domain receptors have demonstrated antibacterial activity [38]. CLEC-4, a C-type lectin-like domain receptor, has been tied to nutrient sensing through DAF-12, AAK-2, and CRH-1 [38]. These findings relate the bacterial diet to innate immunity in the worm [38]. On the other hand, the *C. elegans* genome also encodes over 1,000 G-protein coupled receptors, many of which are orphan receptors, meaning their ligands are unknown, some of which could have microbial ligands [37,39]. This arsenal of GPCRs allows for *C. elegans* to tailor their innate immunity response [37]. The epidermal dihydrocaffeic acid receptor 1 (DCAR-1) is expressed during wound healing from fungal infection. Through PMK-1, activation of DCAR-1 leads to the expression of antimicrobial peptides and initiation of the healing event [37]. GPCRs can also respond to pathogenic bacteria. For instance, *C. elegans* senses the *Pseudomonas aeruginosa* small molecules phenazine-1-carboxamide and pyochelin, through sensory neuron GPCRs, leading to pathogen avoidance [37]. Interestingly, neuronal GPCRs are key regulators of innate immunity in the worm, such as neuropeptide receptor-1 (NPR-1) which responds to *P. aeruginosa* by inducing PMK-1 [40]. GPCRs expressed in the intestine are also involved in the response to microbes. For example, an ortholog of follicle stimulating hormone (FSH-1) responds to Gram-positive and Gram-negative pathogens [40]. This receptor, which signals through PMK-1, is also involved in responding to oxidative stress and heavy metal toxicity. Altogether, these findings demonstrate the multifunctionality and promiscuity of *C. elegans* GPCRs [40].

## 3. The UPR^mt^ and Innate Immunity 

The UPR^mt^ is an additional inducer of innate immunity in the nematode. The UPR^mt^ senses unfolded proteins in the mitochondrion, as opposed to the cytosolic unfolded protein response (UPR^cyt^), which senses unfolded proteins in the cytosol [41]. The UPR^mt^ is activated during mitochondrial stress, leading to resolution of the stress by degrading damaged proteins or through mitophagy (removal of damaged mitochondria) [42] (Figure 2). Transcriptional activation in response to the mitochondrial signal, mediated by the transcription factor ATFS-1, also leads to increased expression of innate immunity effectors, such as upregulation of C-type lectins and antimicrobial peptide expression [43] (Figure 2). Indeed, an increase in ROS or unfolded proteins in the mitochondrion decreases mitochondrial import efficiency, resulting in ATFS-1 accumulation in the cytosol. Due to its nuclear targeting sequence, ATFS-1 is transported to the nucleus rather than being degraded. After reaching nuclear localization, the transcription factor activates a series of genes to restore mitochondrial function [44]. 

Bacterial pathogens are known to secrete toxins that disrupt mitochondrial function to aid in their proliferation. For instance, *Streptomyces* produces antimycin and oligomycin which are both inhibitors of cytochrome c reductase and ATP synthase, respectively [45]. This will activate UPR^mt^, ROS production, and inflammasome activation [46,47]. It has been shown that *P. aeruginosa* PA14 can target UPR^mt^ in *C. elegans* by catabolizing leucine and valine, which helps it to evade innate immunity [48]. In brief, *P. aeruginosa* restricts branched chain amino acids through the *Pseudomonas* enzyme FadE2, which decreases ATP production for signaling through UPR^mt^, leading to bacterial persistence in the worm gut and slow killing (as opposed to fast killing which is toxin-based) [48]. Recently, *Salmonella enterica* was reported to target ATF5 (an ATFS-1 homolog) to establish infection in mice [49]. ATF5 counteracts infection by *Salmonella* by inducing innate immunity and the satiety response to prevent any further ingestion of pathogenic bacteria [49]. 

## 4. The UPR^mt^ in Longevity and Fitness

A landmark report by Han et al. demonstrated that some mutant *E. coli* strains that induce UPR^mt^, through most notably the extracellular polysaccharide colanic acid, were shown to extend lifespan and promote rescue phenotypes for Tau protein aggregation and somatic cell overgrowth in *C. elegans* [50]. Further, a paper by Govindan et al. demonstrated an interplay between the reactive oxygen species (ROS) produced by *E. coli* OP50 that interact with *C. elegans* mitochondria by increasing translation from the hsp-6p::GFP and hsp-60p::GFP stable constructs, which monitor UPR^mt^ activation [51]. The ROS produced by the bacterial diet had some effect on normal worm development and was later determined to be ATFS-1-dependent [51]. In a paper by Haçariz et al., native microbiome members *Comamonas* sp. MYb 131 and *Chryseobacterium* sp. CHNTR MYb 120 both upregulated transcription of hsp-16.2, a marker for UPR^cyt^. However, gene marker expression for UPR^mt^ (i.e., *hsp-6* and *hsp-60*) were found to be insignificant compared to the *E. coli* OP50 standard laboratory diet [14]. This might be due to the stage at which these worms were collected for transcriptomics analysis (L4, young adult) or that these members of the microbiome do not target UPR^mt^, such as *Pseudomonas* and *Salmonella* species do in *C. elegans* and mice, respectively [48,49,52].

## 5. *C. elegans* Models of Parkinson’s Disease

Recent reports have demonstrated that *C. elegans* with PD-like phenotypes had decreased ATP production in response to mutations in PINK1 (PARK6) and Parkin (PARK2) [53]. PINK1 and Parkin regulate mitophagy in the worm. Under normal conditions, PINK1 is transported into the mitochondria and is degraded. However, under mitochondrial stress, PINK1 is unable to cross the membrane and becomes embedded in the mitochondrial outer membrane. Parkin is then recruited from the cytosol to the outer membrane where PINK1 activates the ubiquitin ligase activity of Parkin through phosphorylation. PINK1 also phosphorylates free ubiquitin, leading to Parkin-dependent ubiquitination of mitochondrial proteins, with the goal of recruiting autophagosome machinery to dispose of the damaged mitochondria [42]. Mutations in PINK1 or Parkin will lead to a disruption in deficient mitochondrial clearance, resulting in neuronal loss. In these cases, PD pathology results from the inefficiency of UPR^mt^. However, genetic mutations are responsible for 5–10% of PD cases [54]. The role of the microbiome in PD remains elusive. In *C. elegans* models of PD, bacterial ligands from colonizing bacteria, such as with *Lactobacillus paracasei* HII01, were shown to negate PD molecular effectors, such as reduced aggregation of α-synuclein and formation of Lewy bodies [55]. Interestingly, metformin was shown to rescue a PD-like paralysis phenotype in *bcat-1* mutant worms defective in branched-chain amino acid metabolism, which have hyperactive mitochondria. The drug decreased mitochondrial respiration through inhibition of Complex I, further indicating that PD symptoms are of mitochondrial origin [56]. 

## 6. The Kynurenine Pathway and Innate Immunity in *C. elegans*

In mammals, L-tryptophan is catabolized primarily through the kynurenine pathway. The activity of the kynurenine pathway is linked to activation of the aryl hydrocarbon receptor (AhR), which mediates development and influences immune function [57]. L-kynurenine itself is a ligand of AhR [57]. L-kynurenine binding to AhR is related to immunosuppression seen in aging humans and mammals [58]. However, L-kynurenine does not bind to the AhR homolog in the worm, suggesting that this small molecule may act differently in *C. elegans*. In *C. elegans*, the kynurenine pathway has been linked to nervous system development [57], and its role in innate immunity is not well characterized. Recently, it was observed that a kynurenine pathway metabolite, 3-hydroxyanthranilic acid, was found to activate innate immunity against a *Pseudomonas aeruginosa* bacterial pathogen in *C. elegans*, which induced the kynurenine pathway to a higher level compared to the less pathogenic bacterial species of *E. coli* [59,60]. The inhibition of 3-hydroxyanthranilate 3,4-dioxygenase (the enzyme that metabolizes 3-hydroxyanthranilic acid) in the worm, such as in the *haao-1* mutant, lead to the same outcome [59]. The effect of 3-hydroxyanthranilic acid was found to be direct on the pathogen itself, and not an indirect signaling effect through immune pathways [59]. 

## 7. α-Synuclein, UPR^mt^ and the Kynurenine Pathway

Numerous reports have stated that α-synuclein aggregation preferentially interacts and accumulates in mitochondria of neurons in PD, more specifically in substantia nigra compacta neurons, which inhibits mitochondrial function and induces UPR^mt^ [61,62,63]. This leads to UPR^mt^ mitigating the effects of α-synuclein, but over time, the response subsides and leads to mitophagy [63]. Mitophagy entails ATP production inhibition and neuronal cell death [42]. In both mammals and worms, a mitochondrial outer membrane enzyme, kynurenine monooxygenase (KMO), converts L-tryptophan and L-kynurenine into 3-hydroxykynurenine, which is neurotoxic [64]. KMO is part of the kynurenine pathway (KP), responsible for the committed step [65]. Interestingly, knockdown or allelic mutation of cinnabar, the KMO homolog in *Drosophila melanogaster*, leads to mitochondrial morphological changes independent of relative KP metabolite levels [64].

## 8. Microbiome and the Kynurenine Pathway: Initiators and a Potentiators of α-Synuclein Aggregation

The role of the microbiome in the initiation of α-synuclein aggregation is well documented. In humans, changes in gut microbiome composition were observed preceding the aggregation of α-synuclein in the brain [5]. Further, fecal transplants from PD patients to germ-free mice induced α-synuclein aggregation in the gut, which was later found in the brain and led to motor deficits in these animals [5]. In addition, in germ-free mice, short chain fatty acids produced by the gut microbiome increased α-synuclein pathology through microglial cell activation [5]. This study demonstrated that the gut microbiome can potentially act as an initiator of PD pathology through the initiation of α-synuclein aggregation in the gut [5].

The link between the enteric nervous system and the central nervous system is the vagus nerve. One hypothesis of how α-synuclein accumulates in the substantia nigra region of the brain is through the indirect effect of gut dysbiosis. Inflammation caused by gut microbiome dysbiosis leads to a leaky gut, increasing the permeability of the gut to small inflammatory molecules such as LPS [66]. Gut inflammation increases with age due to changes in gut microbiome composition and/or waning of intestinal immunity, hence the term “inflammaging”, which explains the increased incidence of PD with age [67,68]. However, recently, a higher incidence of PD in younger people has been reported, which could be due to exposure to environmental toxins such as pesticides, leading to increased gut breaches [69]. Therefore, gut permeability could be a major factor in the onset of PD. Another plausible theory for the onset of PD could be the microbiome alteration of levels of kynurenine pathway metabolites, which ultimately play a role in the generation of neurotoxic ones such as quinolinic acid [70].

## 9. Microbial Contributions to Kynurenine Pathway Metabolites

L-tryptophan is catabolized through three major pathways: (1) the kynurenine pathway, (2) the serotonin pathway, and (3) the indole pathway [71]. The catabolism of L-tryptophan also results in the generation of nicotinamide adenine dinucleotide (NAD+), an important cofactor related to many neurodegenerative diseases [58]. The indole pathway relates to the direct metabolism of tryptophan by the gut microbiome [65]. The production of indole metabolites and intermediates from tryptophan by the gut microbiome affects host responses to their bacterial symbionts, and the effects of these metabolites are far-reaching. Indole production by *E. coli* was demonstrated to promote fitness and health span in *C. elegans* [72,73]. Its production was found to increase with worm age, and the effect was found to be DAF 16-dependent, which is a FOXO transcription factor [73]. Recently, indole-3-lactic acid produced by *Lactobacillus* sp. was demonstrated to alleviate the gut immune response and modulate the relative proportion of beneficial bacterial phyla in the microbiome in mice, preventing lethal infection by *Citrobacter rodentium* [74]. Hence, activity of the indole pathway seems to be beneficial for the host.

The kynurenine pathway is initiated by the activation of either indoleamine-2,3-dioxygenase (IDO) or tryptophan 2,3-dioxygenase (TDO) in mammals [65]. In humans, the kynurenine/tryptophan ratio in plasma is used as a biomarker for immune system activation by the activity of indoleamine 2,3-dioxygenase (IDO) [75]. A low ratio reflects a systemic immunosuppression due to L-tryptophan not being shuttled into the kynurenine pathway, whereas a high ratio is indicative of immune activation due to the activity of kynurenine (and intermediates) in the immune cell response [75]. The microbiome indirectly affects this ratio by altering the pool of indole and tryptophan metabolites. Moreover, the crosstalk between the gut microbiome and the innate immune system affects levels of inflammatory mediators involved in activating the kynurenine pathway. The innate immune system can communicate with microbes though the interaction between PAMPs and TLRs, and activation of TLRs by LPS can initialize the kynurenine pathway [65,76]. Hence, dysbiosis or pathogenic infection would lead to an increase in TLR activation and an increase in inflammation. Consequently, the kynurenine pathway would be activated as well. This suggests that the microbiome and the immune system are intertwined partially through the kynurenine pathway. In addition, the bacterial species comprising the microbiome can partly modulate the immune system by indirectly modulating kynurenine pathway.

## 10. Using Metabolomics and Isotope Labeling to Study Host–Microbiome Interactions

Host–microbiome interactions are complex and challenging to study. While transcriptomics and proteomics can help elucidate the mechanism behind these interactions, metabolomics offers a more dynamic approach, providing a deeper understanding of how the microbiome affects host metabolism and vice versa [77,78]. Metabolomics has been used with model organisms, human samples, and cell lines to help identify metabolic pathways associated with health and disease [79]. Model organisms such as *C. elegans* and *D. melanogaster* are particularly useful to study the effects of a monoxenic microbiome on the host metabolism. For example, untargeted metabolomics has revealed that certain bacterial species impact host metabolites in *D. melanogaster*, such as purines, affecting aging [80]. Similarly, metabolomic studies have shown that gut microbiome-specific metabolites impact the development and physiology of *C. elegans* [81]. Metabolomics gives a big picture of the differences in metabolic profiles between biological conditions. It does not provide direct information about the metabolic pathway activities. Metabolite concentrations are influenced by both production and consumption by various metabolic pathways, so changes in metabolite levels cannot easily be linked with a single pathway [82,83,84]. Stable isotope labeling offers opportunities to overcome these limitations to help identify active pathways [85,86]. Recent studies in *Drosophila* have successfully coupled untargeted metabolomics with isotope labeling, where flies were fed an isotope-labeled food source, ^13^C6-glucose, leading to the discovery of metabolic rewiring during aging [87]. In *C. elegans*, the microbiome diet can be labelled with ^13^C glucose and then fed to the worms, which has led to the discovery that *Chryseobacterium* sp. CHNTR MYb 120 induces trehalose accumulation in the aging worm, compared to *E. coli* OP50 [88]. Metabolomics coupled with isotope labeling is well-positioned to advance our understanding of the interaction between the microbiome, UPR^mt^, and the kynurenine pathway, utilizing precursors such as ^13^C glucose, ^15^N-tryptophan, or ^13^C-tryptophan.

## 11. Conclusions and Future Challenges

PD is a multifactorial disease of mitochondrial origin. Activation of UPR^mt^ and its role in alleviating ROS and degrading unfolded proteins in the mitochondrion are crucially important. The immune regulation aspect of UPR^mt^ is also crucial for molding host–microbiome interactions. We hypothesize that some bacterial members of the native microbiome are likely to induce UPR^mt^ at a therapeutic level, which could lead to remediation of α-synuclein aggregation in PD. As shown in Figure 3, the *C. elegans* model is well-positioned to answer many questions along this line and make discoveries in the proposed mechanism. For example, the elucidation of a possible link between TOL-1 and UPR^mt^ could be of interest. Beneficial microbes could activate TOL-1 in a similar way that *E. faecalis* does to promote host bacterial tolerance. This may be consequently linked to UPR^mt^ activation, which is involved in innate immunity. A recent report demonstrated that bacterial muropeptides from peptidoglycan influenced worm development and resistance to oxidative stress [89]. These wild-type muropeptides from *E. coli* were found to enter intestinal mitochondria and led to UPR^mt^ repression [89]. The link between TOL-1, these small muropeptides, and UPR^mt^ remains to be elucidated. Another study by Govindan et al. reports that ROS generated by *E. coli* are necessary for ATFS-1 activation. ROS generation could potentially lead to the upregulation of kynurenine pathway metabolites, resulting in the activation of UPR^mt^. There could also exist a bacterial ligand that elicits ROS production by the bacteria and/or host. Answers to these questions require carefully designed experiments with genetically distinct *C. elegans* strains and mutant bacterial pathogens and microbiome members. How the kynurenine pathway interacts with UPR^mt^ is still to be defined. KMO, found in the outer membrane of the mitochondrion, could potentially link the kynurenine pathway and UPR^mt^, which is partially supported by a recent study that showed mitochondrial morphology changes in a KMO mutant of *D. melanogaster* [54]. Integrating metabolomics with isotope labeling could provide insights into the UPR^mt^ and kynurenine pathway dysregulation in PD.

## Figures and Tables

**Figure 1 biomolecules-14-01370-f001:**
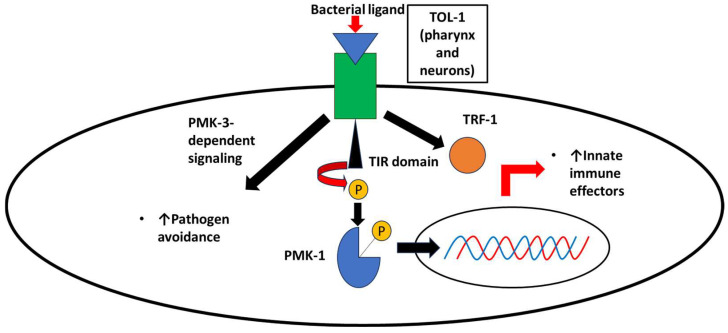
Bacterial ligands interact with TOL-1 in *C. elegans*. Full-length LPS from *S*. Typhimurium, and small muropeptides from peptidoglycan interact with TOL-1 in *C. elegans*, leading to a phosphorylation casacade and upregulation of the gene expression of innate immune effectors, which is PMK-1-dependent. Pathogen avoidance is also upregulated via a PMK-3-dependent signaling mechanism in sensory neurons.

**Figure 2 biomolecules-14-01370-f002:**
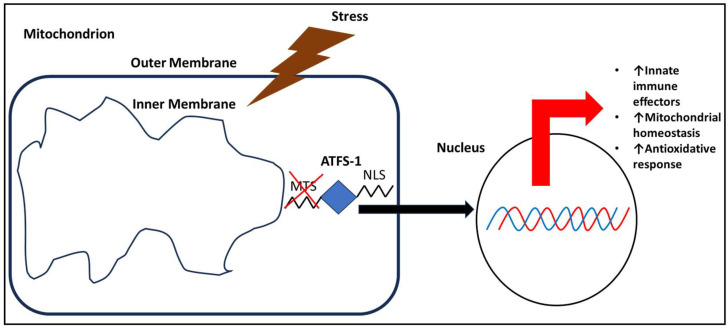
ATFS-1 is localized to the nucleus for gene expression following mitochondrial stress in *C. elegans*. Following induction of the mitochondrial unfolded protein response (UPR^mt^), ATFS-1 directs transcription of genes involved in innate immune effectors, mitochondrial homeostasis, and antioxidative response, promoting longevity.

**Figure 3 biomolecules-14-01370-f003:**
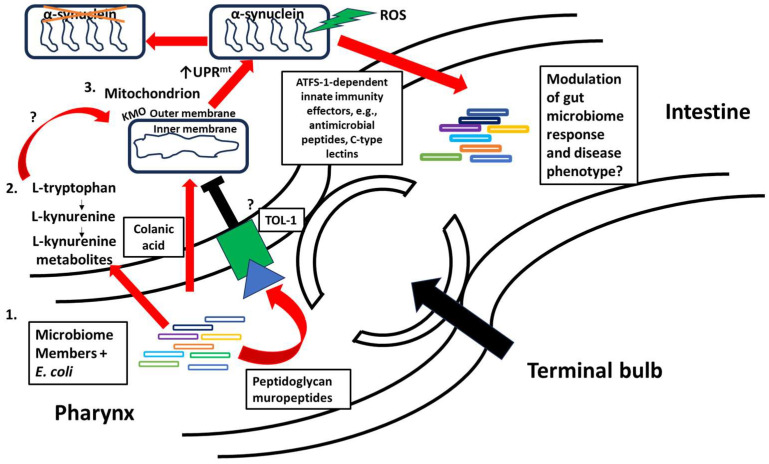
Proposed interactions between the gut microbiome, UPR^mt^, and the kynurenine pathway in *C. elegans*. 1. Microbes, such as *E. coli*, activate or inhibit UPR^mt^, through the extracellular polysaccharide, colanic acid, or small muropeptides from peptidoglycan, respectively. 2. Microbes also influence the kynurenine pathway by interacting with the enzyme kynurenine monooxygenase (KMO), potentially linking mitochondrial homeostasis to the kynurenine pathway. 3. Once UPR^mt^ is induced, α-synuclein aggregation is resolved and the gut microbiome may be modulated, leading to disease phenotype alteration.

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
