# Peer review of "Use of Caenorhabditis elegans to Unravel the Tripartite Interaction of Kynurenine Pathway, UPRmt and Microbiome in Parkinson’s Disease"

_biomolecules, 2024, doi:10.3390/biom14111370_

Round 1

Reviewer 1 Report

Comments and Suggestions for Authors

This well-written Review manuscript presents a detailed overview of the relationship between the gut microbiome and its role in neurodegenerative diseases like Parkinson's in C. elegans as a model organism. It mainly focuses on the tripartite interactions among the mitochondrial unfolded protein response (UPRmt), the kynurenine pathway, and the microbiome.

However, I suggest clarifying a few points to enhance the impact of your review.

It will be beneficial to include details in the introduction about the known role of the gut microbiome in Parkinson's disease in humans.

C. elegans lacks a traditional adaptive immune system. The authors should make the distinction clear, as the present manuscript gives the impression of similarity between the human immune system and that of C. elegans.

Section 4. UPRmt in Longevity and Fitness - it is unclear what its relation is with PD.

line 36: C. elegans has been studied for studying the host microbe interactions well before its microbiome was identified. Authors should make changes to reflect this fact.

Authors may include PMID: 28622510 to demonstrate bacterial mutants' utility in studying pathophysiological functions in C. elegans.

Line 276: The correct reference for this statement would be PMID: 28827345; please cite the correct reference.

The illustration should be improved to clearly represent the pathways in the context of C. elegans. The labeling on the figure needs to be more apparent in indicating the anatomical structures being represented.

For those new to worm biology, the review should briefly highlight the potential or known drawbacks of using C. elegans to study mammalian PD.

Minor: There are numerous instances (too many to list individually) of incorrect formatting of scientific nomenclature; please correct them.

Comments on the Quality of English Language

The quality of the English language is good and overall well-written, with minor formatting errors.

Author Response

Reviewer 1

This well-written Review manuscript presents a detailed overview of the relationship between the gut microbiome and its role in neurodegenerative diseases like Parkinson's in C. elegans as a model organism. It mainly focuses on the tripartite interactions among the mitochondrial unfolded protein response (UPRmt), the kynurenine pathway, and the microbiome.

However, I suggest clarifying a few points to enhance the impact of your review.

Comment #1: It will be beneficial to include details in the introduction about the known role of the gut microbiome in Parkinson's disease in humans.

Response #1: We thank the reviewer for this comment; we have included some information about the role of the gut microbiome in PD in lines 91-105, where we discuss the different genera associated PD, the role of short-chain fatty acids, where we see a decrease in these molecules in PD patients, and the link between PD and gut dysbiosis.

References cited:

  1. Devos, D.; Lebouvier, T.; Lardeux, B.; Biraud, M.; Rouaud, T.; Pouclet, H.; Coron, E.; Bruley des Varannes, S.; Naveilhan, P.; Nguyen, J.M.; et al. Colonic inflammation in Parkinson's disease. Neurobiol Dis 2013, 50, 42-48, doi:10.1016/j.nbd.2012.09.007.
  2. Hirayama, M.; Ohno, K. Parkinson's Disease and Gut Microbiota. Ann Nutr Metab 2021, 77 Suppl 2, 28-35, doi:10.1159/000518147.
  3. Salat-Foix, D.; Tran, K.; Ranawaya, R.; Meddings, J.; Suchowersky, O. Increased intestinal permeability and Parkinson disease patients: chicken or egg? Can J Neurol Sci 2012, 39, 185-188, doi:10.1017/s0317167100013202.
  4. Gallop, A.; Weagley, J.; Paracha, S.U.; Grossberg, G. The Role of The Gut Microbiome in Parkinson's Disease. J Geriatr Psychiatry Neurol 2021, 34, 253-262, doi:10.1177/08919887211018268.
  5. Sorboni, S.G.; Moghaddam, H.S.; Jafarzadeh-Esfehani, R.; Soleimanpour, S. A Comprehensive Review on the Role of the Gut Microbiome in Human Neurological Disorders. Clin Microbiol Rev 2022, 35, e0033820, doi:10.1128/cmr.00338-20.
  6. Chiang, H.L.; Lin, C.H. Altered Gut Microbiome and Intestinal Pathology in Parkinson's Disease. J Mov Disord 2019, 12, 67-83, doi:10.14802/jmd.18067.

Comment #2: C. elegans lacks a traditional adaptive immune system. The authors should make the distinction clear, as the present manuscript gives the impression of similarity between the human immune system and that of C. elegans.

Response #2: We acknowledge the comment made by the reviewer and have added lines 40-50, where we state that C. elegans does not possess an adaptive immune system, however, we discuss about its innate immune system, which comprises a sole TLR homolog, TOL-1, its inducible antimicrobial responses and various signalling pathways that are conserved with humans.

References cited:

  1. Gravato-Nobre, M.J.; Hodgkin, J. Caenorhabditis elegans as a model for innate immunity to pathogens. Cellular Microbiology 2005, 7, 741-751, doi:https://doi.org/10.1111/j.1462-5822.2005.00523.x.
  2. Tenor, J.L.; Aballay, A. A conserved Toll-like receptor is required for Caenorhabditis elegans innate immunity. EMBO reports 2008, 9, 103-109-109, doi:https://doi.org/10.1038/sj.embor.7401104.
  3. Mallo, G.V.; Kurz, C.L.; Couillault, C.; Pujol, N.; Granjeaud, S.; Kohara, Y.; Ewbank, J.J. Inducible antibacterial defense system in C. elegans. Curr Biol 2002, 12, 1209-1214, doi:10.1016/s0960-9822(02)00928-4.

Comment #3: Section 4. UPRmt in Longevity and Fitness - it is unclear what its relation is with PD.

Response #3: We are glad that the reviewer brought this comment to our attention. Longevity and fitness have long been debated in their roles in health and disease. Some genes in C. elegans lead to increased lifespan (longevity), but not necessarily healthspan (fitness), see publication by Bansal et al., 2015 for more information (DOI: 10.1073/pnas.1412192112). As PD is a disease usually associated with aging, theorized to be due to waning of the immune system and increased gut permeability, influencing the microbiome to modulate UPRmt and the expression of genes involved in this process, could decrease the incidence of PD, by increasing restoring the important functions of this unfolded protein response.

Comment #4: line 36: C. elegans has been studied for studying the host microbe interactions well before its microbiome was identified. Authors should make changes to reflect this fact.

Response #4: We thank the reviewer for this comment. In fact, we do reference in lines the landmark paper by Cabreiro et al., 2013 (DOI: 10.1016/j.cell.2013.02.035), where the authors showed between E. coli OP50 metabolism, the standard laboratory diet of C. elegans, the anti-diabetic drug metformin and worm metabolism in lines 55-60.

Comment #5: Authors may include PMID: 28622510 to demonstrate bacterial mutants' utility in studying pathophysiological functions in C. elegans.

Response # 5: We thank the reviewer for this comment, we do reference the report by Han et al., in lines 217-220.

Comment #6: Line 276: The correct reference for this statement would be PMID: 28827345; please cite the correct reference.

Response #6: We acknowledge the report by Sonowal et al. as being the seminal report with regards to indole metabolites influencing the gut microbiome in C. elegans, and have changed the reference accordingly. It is now rewritten as “Indole production by E. coli was demonstrated to promote fitness and healthspan in C. elegans [74,75].” in lines 315-316.

Comment #7: The illustration should be improved to clearly represent the pathways in the context of C. elegans. The labeling on the figure needs to be more apparent in indicating the anatomical structures being represented.

Response #7: We are grateful for the reviewer’s comment and have improved the illustration of the tripartite interaction between the kynurenine pathway, UPRmt and the microbiome. We have also included figures 1 and 2 for completeness.

Comment #8: For those new to worm biology, the review should briefly highlight the potential or known drawbacks of using C. elegans to study mammalian PD.

Response #8: We thank the reviewer for this point, we have alluded to the fact that C. elegans lacks an α-synuclein ortholog and that there is differential neuronal connectivity in this organism, i.e., lack of a central nervous system, in lines 79-81. We cite PMID: 29480229.

Comment #9 : Minor: There are numerous instances (too many to list individually) of incorrect formatting of scientific nomenclature; please correct them.

Response #9: We thank the reviewer for this observation have corrected them (highlighted in red).

Reviewer 2 Report

Comments and Suggestions for Authors

In this review article, the authors summarize the relationship between C. elegans and the gut microbiota and their impact on immunity, mitochondrial unfolded protein response (UPRmt), amino acid metabolism, and parkinsonism. The content is well-organized and covers important publications related to the topic, but additional explanations and figures should be added before publication.

Main points:

Figure 1 is somewhat busy and difficult to understand. The red number 1-3A and 3B may be explained in the figure legend. ROS production should be included in the figure. Mitochondrial matrix and α-synuclein should be distinctly represented.

It is recommended to add figures of TOL-1 signaling from ligand to downstream kinases after section 2, and of UPRmt including ATFS-1 localization from mitochondria to the nucleus and its target gene after section 3 and 4. The original Figure 1 may be divided into these additional figures.

In line 110, the "peptidoglycan muropeptides" and "SagA enzyme" should be briefly explained.

In lines 167 and 168, antimycin and oligomycin can be described as inhibitors of "mitochondrial respiration" or "cytochrome c reductase and ATP synthase, respectively" rather than "important mitochondrial structures".

In line 202, "its ubiquitin ligase activity" may be misleading and can be "ubiquitin ligase activity of Parkin". The next sentence, "PINK1 also activates free ubiquitin" seems unclear because PINK1 phosphorylates ubiquitin which mediate feed-forward loop for mitochondrial protein ubiquitination by Parkin.

Minor points:

Line 143, "G-protein coupled receptors" can be abbreviated as GPCRs is already described.

Line 233, a period in "α-. Synuclein" should be removed.

Some species name are not italicized and should be checked again throughout the manuscript, such as C. elegans in lines 106, 142, and 277, Salmonella enterica in line 174, E. coli in lines 183, and 276, and Drosophila melanogaster in line 242.

Uppercase of UPRmt and UPRcyt should be checked in lines 169, 180, and 189.

In the reference list, the journal information is missing in some citations (e.g. ref 6, 25, 27, and 30~).

Comments on the Quality of English Language

No problem with the English in this manuscript was found except for the minor point described above.

Author Response

Reviewer 2

In this review article, the authors summarize the relationship between C. elegans and the gut microbiota and their impact on immunity, mitochondrial unfolded protein response (UPRmt), amino acid metabolism, and parkinsonism. The content is well-organized and covers important publications related to the topic, but additional explanations and figures should be added before publication.

Main points:

Comment #1: Figure 1 is somewhat busy and difficult to understand. The red number 1-3A and 3B may be explained in the figure legend. ROS production should be included in the figure. Mitochondrial matrix and α-synuclein should be distinctly represented.

Response #1: We appreciate the reviewer’s comment and have amended the figure in question, which is now Figure 3. We have included ROS in the figure, which induces the aggregation of α-synuclein. We have also distinctly separated the locations of the worm structures and cellular organelles.

Comment #2: It is recommended to add figures of TOL-1 signaling from ligand to downstream kinases after section 2, and of UPRmt including ATFS-1 localization from mitochondria to the nucleus and its target gene after section 3 and 4. The original Figure 1 may be divided into these additional figures.

Response #2: To address this valuable comment, we have included Figures 1 and 2 to this review manuscript. In figure 1, we illustrate what is known in terms of downstream kinases following TOL-1 signalling and in Figure 2, we illustrate the transcriptional activation by ATFS-1 in the nucleus following mitochondrial stress.

Comment#3: In line 110, the "peptidoglycan muropeptides" and "SagA enzyme" should be briefly explained.

Response#3: We thank the reviewer for this comment. We have added lines 138-141 to briefly explain this mechanism: “In brief, small muropeptides generated from E. faecalis peptidoglycan by this enzyme, through signalling by TOL-1, allow for S. Typhimurium persistence in the gut, which does not kill the worm prematurely.” On line 138-141.

Comment #4: In lines 167 and 168, antimycin and oligomycin can be described as inhibitors of "mitochondrial respiration" or "cytochrome c reductase and ATP synthase, respectively" rather than "important mitochondrial structures".

Response #4: We thank the reviewer for this comment. We have changed the sentence to “Bacterial pathogens are known to secrete toxins that disrupt mitochondrial function to aid in their proliferation. For instance, Streptomyces produces antimycin and oligomycin which are both inhibitors of cytochrome c reductase and ATP synthase, respectively”., on line 206.

Comment #5: In line 202, "its ubiquitin ligase activity" may be misleading and can be "ubiquitin ligase activity of Parkin". The next sentence, "PINK1 also activates free ubiquitin" seems unclear because PINK1 phosphorylates ubiquitin which mediate feed-forward loop for mitochondrial protein ubiquitination by Parkin.

Response #5: We thank the reviewer for this comment and have changed “its ubiquitin ligase activity” to “ubiquitin ligase activity of Parkin.” We also have changed “PINK1 also activates free ubiquitin” to “PINK1 also phosphorylates free ubiquitin, leading to Parkin-dependent ubiquitination of mitochondrial proteins, with the goal of recruiting autophagosome machinery to dispose of the damaged mitochondria [46].” on line 240-242.

Minor points:

Comment #6: Line 143, "G-protein coupled receptors" can be abbreviated as GPCRs is already described.

Response #6: We have abbreviated G-protein coupled receptors” to GPCRs, now on line 147.

Comment #7: Line 233, a period in "α-. Synuclein" should be removed.

Response #7: The period has been removed.

Comment#8: Some species name are not italicized and should be checked again throughout the manuscript, such as C. elegans in lines 106, 142, and 277, Salmonella enterica in line 174, E. coli in lines 183, and 276, and Drosophila melanogaster in line 242.

Response#8: The instances of “C. elegans”, “Salmonella enterica” and “E. coli” have all been italicized.

Comment #9: Uppercase of UPRmt and UPRcyt should be checked in lines 169, 180, and 189.

Response #9: mt and cyt in UPR have all been made uppercase.

Comment #10: In the reference list, the journal information is missing in some citations (e.g. ref 6, 25, 27, and 30~).

Response #10: They have all been amended to reflect the journal’s requirements.

Round 2

Reviewer 2 Report

Comments and Suggestions for Authors

All points suggested were revised.